# Interventions to improve linkage along the HIV-tuberculosis care cascades in low- and middle-income countries: A systematic review and meta-analysis

Angela Salomon[1,2], Stephanie Law[2], Cheryl Johnson[3], Annabel Baddeley[4], Ajay Rangaraj[3], Satvinder Singh[5], Amrita Daftary[6,7]*

1 School of Medicine, Queen's University, Kingston, Canada, 2 McGill International TB Centre, Research Institute of the McGill University Health Centre, Montréal, Canada, 3 Global HIV, Hepatitis and STI Programmes, World Health Organization, Geneva, Switzerland, 4 Global TB Programme, World Health Organization, Geneva, Switzerland, 5 The Global Fund to Fight AIDS, Tuberculosis and Malaria, Geneva, Switzerland, 6 School of Global Health and Dahdaleh Institute of Global Health Research, York University, Toronto, Canada, 7 Centre for the AIDS Programme of Research in South Africa, University of KwaZulu-Natal, Durban, South Africa

* adaftary@yorku.ca

**Data Availability Statement:** All relevant data are within the manuscript and its Supporting Information files.

## Abstract

### Introduction

In support of global targets to end HIV/AIDS and tuberculosis (TB) by 2030, we reviewed interventions aiming to improve TB case-detection and anti-TB treatment among people living with HIV (PLHIV) and HIV testing and antiretroviral treatment initiation among people with TB disease in low- and middle-income countries (LMICs).

### Methods

We conducted a systematic review of comparative (quasi-)experimental interventional studies published in Medline or EMBASE between January 2003-July 2021. We performed random-effects effect meta-analyses (DerSimonian and Laird method) for interventions that were homogenous (based on intervention descriptions); for others we narratively synthesized the intervention effect. Studies were assessed using ROBINS-I, Cochrane Risk-of-Bias, and GRADE. (PROSPERO #CRD42018109629).

### Results

Of 21,516 retrieved studies, 23 were included, contributing 53 arms and 84,884 participants from 4 continents. Five interventions were analyzed: co-location of test and/or treatment services; patient education and counselling; dedicated personnel; peer support; and financial support. A majority were implemented in primary health facilities (n = 22) and reported on HIV outcomes in people with TB (n = 18). Service co-location had the most consistent positive effect on HIV testing and treatment initiation among people with TB, and TB case-detection among PLHIV. Other interventions were heterogenous, implemented concurrent with

**Funding:** This study was funded by United States Agency for International Development through the World Health Organization. The funders had no role in study design, data collection and analysis, decision to publish, or preparation of the manuscript.

**Competing interests:** We have read the journal's policy and the authors of this manuscript have the following competing interests: AD is a member of the Editorial Board of PLOS ONE. This does not alter our adherence to PLOS ONE policies on sharing data and materials. The remaining authors have declared that no competing interests exist. The opinions expressed in the manuscript are those of the authors alone. They do not purport to reflect the opinions or views of the World Health Organization, or The Global Fund to Fight AIDS, Tuberculosis and Malaria, or its members.

standard-of-care strategies and/or diverse facility-level improvements, and produced mixed effects. Operational system, human resource, and/or laboratory strengthening were common within successful interventions. Most studies had a moderate to serious risk of bias.

## Conclusions

This review provides operational clarity on intervention models that can support early linkages between the TB and HIV care cascades. The findings have supported the World Health Organization 2020 HIV Service Delivery Guidelines update. Further research is needed to evaluate the distinct effect of education and counselling, financial support, and dedicated personnel interventions, and to explore the role of community-based, virtual, and differentiated service delivery models in addressing TB-HIV co-morbidity.

## Introduction

HIV and tuberculosis (TB) are inextricably linked [1]. TB is the leading opportunistic infection among people living with HIV (PLHIV), responsible for approximately 30% of all AIDS-related deaths. HIV, through weakening of the immune system, is the leading risk factor for development of TB disease in people with TB infection, and contributes to 15% of TB-related deaths [2]. The World Health Organization (WHO) recommends offering routine HIV testing to all patients with presumptive and diagnosed TB, routine TB screening for TB symptoms of all PLHIV, and starting all patients with TB and HIV on both anti-retroviral therapy and anti-TB treatment (ATT) [3,4]. These strategies have helped to reduce morbidity and mortality from HIV-associated TB, saving an estimated 7.3 million lives since 2005.[2, 3] Nevertheless, significant gaps remain in the TB-HIV care cascade particularly the detection of co-morbidity, and subsequent linkage to treatment. In 2019, 31% of people with TB remained unaware of their HIV status; fewer than half of those estimated to have HIV coinfection received ART. Among PLHIV, an estimated 44% of TB remained undetected and, therefore, untreated [2,5]. Resource-limited settings that face dual burdens of disease continue to report the worst outcomes in TB-HIV [2,5].

Gaps in linkage between the HIV and TB care cascades may be partly explained by the inadequate adoption and implementation of global recommendations within country programs, slow scale-up of new technologies, particularly rapid TB diagnostics, and disparate funding, monitoring and evaluation systems for HIV and TB [5]. Programmatic guidance for the integration of TB and HIV services such as successful models and implementation considerations is also limited. Published reviews have focused on the effect of specific interventions such as patient food support, workplace programs, and private-public partnerships, and focused on prevention of TB disease in PLHIV, and adherence to ART and/or ATT in those receiving dual treatment [6–11]. This systematic review uniquely assesses the full spectrum of non-clinical interventions targeted to patients, providers and programs in low- and middle-income countries (LMICs), and focusses on two critical underexplored outcomes in the TB-HIV care cascade–testing and diagnosis of TB or HIV co-morbidity in people with one known infection, and subsequent linkage to its treatment. Our overarching goal was to inform the WHO 2020 HIV Service Delivery Guidelines update.

## Methods

This multi-method systematic review and meta-analysis adhered to PRISMA (Preferred Reporting Items for Systematic Reviews and Meta-Analyses) [12] and SWiM (Synthesis

Without Meta-Analysis) [13] reporting guidelines (S1 File: PRISMA Checklist). The review protocol is registered with the International Prospective Register of Systematic Reviews (PROSPERO; Reg# CRD42018109629 –S2 File: Systematic review protocol). This review did not require ethics review.

### Research questions

The systematic review focused on two questions:

- In patients with TB disease in LMICs, what interventions improve HIV testing and linkage to ART (PICO 1).

- In people living with HIV in LMICs, what interventions improve TB case-detection and linkage to ATT (PICO 2).

### Search strategy and selection criteria

With the assistance of medical librarians, we searched three electronic databases (Medline (OVID), Embase, Embase Classic) for peer-reviewed articles and conference abstracts published between 1 Jan 2003 and 9 July 2021, three conference abstract databases (International AIDS Society, Conference on Retroviruses and Opportunistic Infections, and Union World Conference on Lung Health) in 2017 and 2018, and reference lists from included studies. The search strategy used the following key terms and their appropriate synonyms: 1) tuberculosis, AND 2) human immunodeficiency virus (HIV), AND 3) diagnosis or detection or screening or testing, or referral or linkage or coordination or integration, or treatment initiation, AND 4) low and middle-income countries. The full search strategy can be found in S3 File: Search Strategy.

We included primary observational, quasi-experimental and experimental (randomized controlled) studies that: 1) examined the effect of a patient-, provider-, or health system-level intervention; 2) implemented the intervention in people known to have HIV, TB disease, or both; 3) reported on one or more of our primary outcomes (Table 1); 4) had at least two study arms (e.g. a control and intervention arm); and 5) was conducted in an LMIC (GNI per capita <12,695 USD per year, as defined by the World Bank) [14]. Secondary outcomes included mortality rate, time to ART initiation, and time to ATT initiation. We excluded studies examining surgical, biomedical, or diagnostic tools/algorithms; secondary analyses/reviews, commentaries, editorials, case reports (<10 participants) and qualitative studies; non-English studies; and studies with interventions implemented at a "population-level" (i.e., not explicitly

**Table 1. Outcomes of the systematic review.**

| Primary outcomes |
| --- |
| **TB case-detection** (participants diagnosed with active TB among participants with HIV) |
| **ATT initiation** (participants initiated on ATT among eligible participants with both HIV and active TB) |
| **HIV testing** (participants tested for HIV among participants with active TB) |
| **ART initiation** (participants initiated on ART among eligible participants with both HIV and active TB) [a] |
| **Secondary outcomes** |
| **Time to ART initiation** |
| **Time to TB treatment initiation** |
| **HIV/TB related mortality** |

PLHIV, People living with HIV; TB, Tuberculosis; ATT, Anti-TB treatment, ART, Antiretroviral therapy.

[a] We included in the denominator only those reported as eligible for initiating ART based on local guidelines at the time of each study [**11**].

among PLHIV or active TB), such as mass/community HIV testing or TB screening and testing campaigns. We did not apply age or other demographic restrictions.

## Data extraction and analysis

Two reviewers (AS, SL) screened all titles and abstracts, followed by full reports of potentially relevant studies; discrepancies were resolved with a third reviewer (AD or VS). Two reviewers (AS, SL) extracted the following data into MS Excel (Microsoft Corporation, Redmond, Washington): primary and secondary outcomes, setting, participant characteristics, standard of care, study interventions, funding sources, and indicators of quality. Authors were systematically contacted for further information when necessary. For included studies, three reviewers (AS, SL, AD) determined primary and secondary outcomes that were potentially affected by the study intervention (i.e., occurred downstream of, and plausibly linked to, the intervention), and estimated unadjusted risk ratios (RRs) and 95% confidence intervals accordingly. Where reported, we used adjusted risk ratios (aRRs) or adjusted hazard ratios (aHRs). We used forest plots to display all effect estimates, including pooled estimates where appropriate, according to the four primary outcomes, and estimated statistical heterogeneity ($I^2$). All statistical analyses were performed using STATA 15.1 (StataCorp, College Station, Texas).

We performed random-effects meta-analyses (DerSimonian and Laird method) for studies implementing interventions that could be pooled (co-location interventions only). For remaining studies which had a high-level of heterogeneity (based on intervention/s, study setting, and populations), we performed a narrative synthesis of the intervention impact. Here, a "narrative synthesis" refers to "an approach to the systematic review and synthesis of findings from multiple studies that relies primarily on the use of words and text to summarise and explain the findings of the synthesis" [15]. Interventions that involved co-location and a second intervention (such as patient education or dedicated personnel) were classified as co-location interventions only. Interventions that involved more than one non-co-location intervention (such as patient education and dedicated personnel) were classified as both. Interventions involving current WHO standard of care strategies, including task-shifting services from specialized to less specialized workers, systematic HIV testing using provider-initiated, opt-out approaches, systematic TB screening and/or testing using standardized tools, health care worker training in TB-HIV care, as well as facility-specific operational improvements [4,16–18] were excluded from analysis, though we note their inclusion in intervention and/or comparator arms. We also performed a narrative synthesis of implementation facilitators and barriers based on the primary outcome(s) affected by interventions, and cost and resource considerations [13].

## Quality assessment

Two reviewers (AS, SL) assessed quality of all included studies using the Risk of Bias in Non-randomized Studies of Interventions (ROBINS-I) [19] tool for non-randomized and non-comparative studies, and the Cochrane Risk-of-Bias tool for randomized trials (cluster or individual) [20] (S4 File: Quality Assessments); we did not exclude studies based on quality. We also developed GRADE evidence profiles for all interventions, pooled and non-pooled [21] (S5 File: Grade Evidence Profiles).

## Results

### Overview

The search strategy identified 21,516 unique studies; 23 studies were eligible, contributing 53 study arms (23 standard of care and 30 intervention) and 84,884 participants (Fig 1). Studies

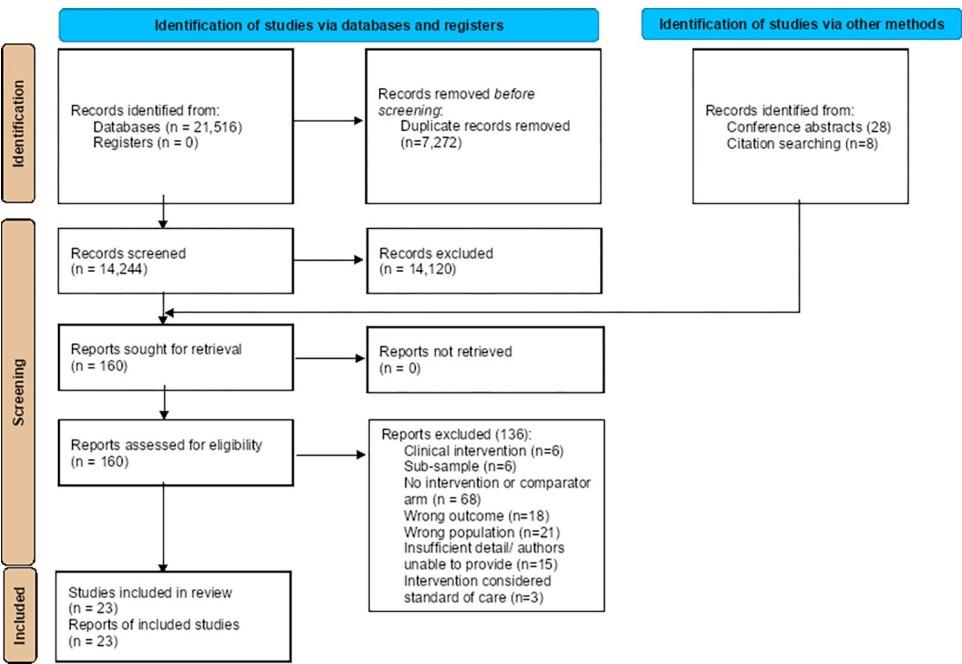

**Fig 1. PRISMA study selection flow chart.** PRISMA = Preferred Reporting Items for Systematic Reviews and Meta-Analyses.

were implemented in LMICs in four WHO regions: Africa (n = 17), South-East Asia (n = 2), Europe (n = 2), and the Americas (n = 2). One study from Peru [22] was implemented in a community setting; remaining studies were in primary health facilities (i.e., hospitals, clinics). Eight studies were limited to adolescents and adults (≥ 12 years), one excluded infants ≤18 months, and three excluded prisoners; others had no demographic-based exclusions. No studies disaggregated results by age or other demographics. Primary outcomes were reported with the following frequencies (Table 2): 1) HIV testing = 7 studies; 2) ART initiation = 16 studies; 3) TB case-detection = 5 studies; and 4) ATT initiation = 2 studies. Six studies reported on more than one outcome. Six studies reported on secondary outcomes (S6 File: Secondary Outcomes). Two studies were randomized controlled trials (RCTs) [23,24]; others were observational.

We analyzed five categories of interventions (Table 3) across 23 studies, including: 1) Co-location (n = 13) of screening, testing and/or treatment services for TB and HIV, at the same facility and/or by the same provider; 2) Patient education and counselling (n = 6) on TB-HIV coinfection; 3) Dedicated personnel (n = 4) to support TB-HIV service delivery; 4) Patient peer support (n = 2) to support TB-HIV service delivery; and 5) Patient financial support (n = 1) (S7 File: Interventions identified in control and intervention arms).

Three studies implemented a single intervention [26,30,38]. Three studies had two distinct interventions [22,36,42]. Eighteen studies implemented a standard of care (SOC) strategy alongside an analyzable intervention [23–25,27–29,31–37,39–41,43,44] (Table 4); effect of the SOC strategy was not analyzed but its concurrent implementation was considered in GRADE certainty assessments.

## Co-location

For PICO 1, we identified eight combinations of co-location, ranging from only co-locating HIV testing with TB services at the same facility, to co-located testing and treatment of HIV

**Table 2. Summary of included studies by outcome.**

PICO 1: HIV TESTING (population = persons with active TB)

| Ref # | Author, year (arm)[a] | Sample size | Study period (years) | Study Population | Country | Setting | Study Design | Intervention (summary)[b] | Outcome RR (95% CI) | Risk of Bias |
|---|---|---|---|---|---|---|---|---|---|---|
| [25] | Agarwal, 2018 (control) | 275 | 2012–2015 | All patients >18 years attending AIDS/TB facilities offering inpatient intensive treatment within 6 distinct oblasts | Ukraine | TB clinics/ HIV clinics (standalone) | Non-randomized, cluster-controlled before-and-after study | Facility-level co-located and systematic HIV testing (for PWTB) and systematic TB screening for (PLHIV); HCW training in caring for co-infected patients; major operational improvements including development of electronic data-management system and other capacity building initiatives to institutionalize best practices in TB-HIV care **Co-location (F–ST only)** + HCW train, Oper Improv, Syst HIV T, Syst TB ST | 1.2 (0.94,1.52)[c] | Moderate |
| | Agarwal, 2018 (Intervention) | 317 | | | | | | | | |
| [26] | Ansa, 2014 (control) | 251 | 2007–2008 | All TB patients (new or previously diagnosed, including transferred cases) registered at a participating facility | Ghana | Hospitals | Non-randomized, cluster-controlled | Control: Referral (no co-location) between TB/HIV services Int 1: Facility-level co-located HIV testing Int 2: provider-level co-located HIV testing, and ART initiation **Co-location (F, P–ST and Tx)** | 1 vs. Control: 1.26 (1.15, 1.39) 2 vs. Control: 1.36 (1.26, 1.47) | Serious |
| | Ansa, 2014 (Intervention 1) | 132 | 2007–2009 | | | | | | | |
| | Ansa, 2014 (Intervention 2) | 207 | 2007–2010 | | | | | | | |
| [27] | Chukwuka, 2011 (control) | 296 | 2008 | All TB patients registered at the study facility | Nigeria | TB clinic (within hospital) | Quasi-experimental (historical control) | Systematic HIV testing (for PWTB) conducted by dedicated personnel (HCT counsellor) posted permanently to the TB centre. **Dedic Person** + Syst HIV T | 2.73 (2.33, 3.22) | Not enough information |
| | Chukwuka, 2011 (Intervention) | 258 | 2009 | | | | | | | |
| [28] | Mwinga, 2008 (control) | 1222 | 2004–2005 | All TB patients registered at a participating facility | Zambia | Hospitals/ clinics | Non-randomized, cluster-controlled | Control: Referral (no co-location) between HTB/HIV services Int 1: Facility-level co-located HIV testing (for PWTB) Int 2: Systematic and provider-level co-located HIV testing (for PWTB); HCW training to deliver as part of routine clinical care. **Co-location (F, P–ST only)** + HCW Train, Syst HIV T | 1 vs. Control 2.29 (2.03,2.57) 2 vs. Control 3.48 (3.12, 3.89) | Serious |
| | Mwinga, 2008 (Intervention 1) | 1589 | 2005 | | | | | | | |
| | Mwinga, 2008 (Intervention 2) | 1337 | 2006 | | | | | | | |
| [29] | Nateniyom, 2008 (control) | 495 | 2006 | All newly diagnosed TB patients (excluding prisoners) registered at a participating facility | Thailand | TB clinics (within hospitals) | Quasi-experimental (historical control) | Systematic and facility-level co-located HIV counselling and testing for PWTB; facilitated through HCW training of nurses and social workers; minor operational improvements including additional meetings and technical support from regional and national TB-HIV administrators and feedback reports; facility-level co-located treatment initiation **Co-location (F–ST and Tx)** + HCW Train, Oper Improv, Syst HIV T | 1.78 (1.63, 1.94) | Moderate |
| | Nateniyom, 2008 (Intervention) | 1000 | 2006 | | | | | | | |
| [22] | Rocha, 2011 (control) | 72 | 2003–2007 | TB patients and their household contacts living in eight shantytowns of norther Lima | Peru | Community | Quasi-experimental (historical control) | Community-level socio-economic activities: patient education and psychological counselling to overcome barriers to TB-diagnosis, treatment, and HIV testing; patient financial training including community-mobilization workshops for income-generation, microenterprise and vocational training; poverty reduction activities involving food and cash transfers **Educ/Couns + Financ Supp** | 3.17 (2.24, 4.49) | Serious |
| | Rocha, 2011 (Intervention) | 318 | 2007–2010 | | | | | | | |

(Continued)

**Table 2.** (Continued)

**PICO 1: HIV TESTING (population = persons with active TB)**

| Ref | Author, year (arm) | Study period (years) | Sample size | Study Population | Country | Setting | Study Design | Intervention (summary) | Outcome RR (95% CI) | Risk of Bias |
|---|---|---|---|---|---|---|---|---|---|---|
| [30] | Van Rie, 2008 (control) | 2004–2005 | 321 | All TB patients >18 months without prior HIV diagnosis registered at participating facility | DRC | TB Clinics (within PHFs) | Non-randomized, cluster-controlled | Control: Referral (no co-location) between TB/HIV services. Int 1: Facility-level co-located HIV testing (for PWTB). Int 2: Provider-level co-located HIV testing (for PWTB) **Co-location (F, P–ST only)** | Arm 1 vs. 2 1.38 (1.28, 1.50) Arm 1 vs. 3 1.43 (1.32, 1.54) | Serious |
|  | Van Rie, 2008 (Intervention 1) |  | 308 |  |  |  |  |  |  |  |
|  | Van Rie, 2008 (Intervention 2) |  | 558 |  |  |  |  |  |  |  |

**PICO 1: ART INITIATION (population = persons with known HIV and active TB)**

| Ref | Author, year (arm) | Study period (years) | Sample size | Study Population | Country | Setting | Study Design | Intervention (summary) | Outcome RR (95% CI) | Risk of Bias |
|---|---|---|---|---|---|---|---|---|---|---|
| [25] | Agarwal, 2018 (control, HIV clinic) | 2012–2015 | 297 | All patients >18 years attending AIDS/TB facilities offering inpatient intensive treatment within 6 distinct oblasts | Ukraine | TB clinics/ HIV clinics (standalone) | Non-randomized, cluster-controlled before-and-after study | Facility-level co-located and systematic HIV testing (for PWTB) and systematic TB screening for (PLHIV); HCW training in caring for co-infected patients; major operational improvements including development of electronic data-management system and other capacity building initiatives to institutionalize best practices in TB-HIV care **Co-location (F–ST only)** + HCW Train, Oper Improv, Syst HIV T, Syst TB ST | HIV clinics 1.49 (1.10, 2.01) [c] TB clinics 2.91 (2.1, 4.04) [c] | Moderate |
|  | Agarwal, 2018 (control, TB clinic) |  | 105 |  |  |  |  |  |  |  |
|  | Agarwal, 2018 (Intervention, HIV clinic) |  | 565 |  |  |  |  |  |  |  |
|  | Agarwal, 2018 (Intervention, TB clinic) |  | 221 |  |  |  |  |  |  |  |
| [26] | Ansa, 2014 (control) | 2007–2008 | 65 | All TB patients (new or previously diagnosed, including transferred cases) registered at a participating facility | Ghana | Hospitals | Non-randomized, cluster-controlled | Control: Referral (no co-location) between TB/HIV services. Int 1: Facility-level co-located HIV testing **Co-location (F–ST only)** | 5.52 (2.68, 11.38) | Serious |
|  | Ansa, 2014 (Intervention 1) | 2007–2009 | 79 |  |  |  |  |  |  |  |
| [27] | Chukwuka, 2011 (control) | 2008 | 56 | All TB patients registered at the study facility | Nigeria | TB clinic (within hospital) | Quasi-experimental (historical control) | Systematic HIV testing (for PWTB) conducted by dedicated personnel (HCT counsellor) posted permanently to the TB centre. **Dedic Person** + Syst HIV T | 2.19 (0.86, 5.57) | Not enough information |
|  | Chukwuka, 2011 (Intervention) | 2009 | 92 |  |  |  |  |  |  |  |
| [31] | Courtenay-Quirk, 2018 (control) | 2013 | 89 | All TB patients newly diagnosed with HIV at participating facility | Tanzania | TB Clinics (standalone) | Modified stepped-wedge design with historical control | Minor operational improvements through addition of HIV testing service register + referral logbooks with fields to facilitate documentation of linkage to care to ART, plus HCW and peer volunteer training on linkage to care and use of the tools. **Peer Supp** + HCW Train, Oper Improv | 0.94 (0.85, 1.03) | Serious |
|  | Courtenay-Quirk, 2018 (Intervention 2) | 2014 | 79 |  |  |  |  |  |  |  |
| [32] | Herce, 2018 (control, clinic A) | 2010–2011 | 131 | All TB/HIV co-infected patients initiating anti-TB treatment, not yet on ARTs and not transferred in from another facility | Zambia | TB Clinics (within PHFs) | Quasi-experimental (historical control) | HCW training and mentorship; systematic and provider-level co-located HIV testing (for PWTB); provider-level co-located ART initiation; major operational improvements including dedicated ART clinic days and synchronized TB and HIV patient follow-up by dedicated TB-HIV personnel; peer-led patient education talks **Co-location (F, P–ST and Tx)** + Educ/Couns, Dedic Person, Peer Supp + HCW Train, Oper Improv, Syst HIV T | Clinic A 1.42 (1.12, 1.81) Clinic B 1.24 (0.92, 1.68) | Low |
|  | Herce, 2018 (control, clinic B) | 2010–2011 | 117 |  |  |  |  |  |  |  |
|  | Herce, 2018 (Intervention, clinic A) | 2011–2012 | 77 |  |  |  |  |  |  |  |
|  | Herce, 2018 (Intervention, clinic B) | 2011–2012 | 148 |  |  |  |  |  |  |  |

*(Continued)*

**Table 2.** (Continued)

PICO 1: HIV TESTING (population = persons with active TB)

| Ref | Study | n | Year | Population | Country | Setting | Design | Intervention | Effect (95% CI) | Risk of bias |
|---|---|---|---|---|---|---|---|---|---|---|
| [33] | Hermans SM, 2012 (control) | 243 | 2007 | All TB/HIV co-infected patients newly initiating TB treatment at the participating facility | Uganda | HIV clinic (standalone) | Quasi-experimental (historical control) | Provider-level co-located HIV testing (for PWTB) delivered by trained, dedicated personnel (peer supporters/lay HCWs); facility-level co-located treatment; major operational improvements including discussion of "difficult cases" at weekly team meetings, placement of ART initiation guides in clinic files, and phone-tracing to prevent loss to follow-up **Co-location (F–ST and Tx; P–Tx only)** + Dedic Person, Peer Supp + HCW Train, Oper Improv | 0.86 (0.74, 0.99) | Moderate |
|  | Hermans SM, 2012 (Intervention) | 229 | 2009 |  |  |  |  |  |  |  |
| [34] | Huerga (control) | 198 | 2005–2007 | All TB patients newly registered at the participating hospital | Kenya | TB Clinic (within hospital) | Quasi-experimental (historical control) | Facility-level co-located (non-systematic) HIV testing and ART initiation at the TB clinic, delivered by three additional dedicated personnel (clinical officer, nurse and counsellor); patient education on HIV prevention **Co-location (F–ST and Tx)** + Educ/Couns, Dedic Person | 5.41 (3.74, 7.82) | Serious |
|  | Huerga (Intervention) | 211 |  |  |  |  |  |  |  |  |
| [35] | Ikeda, 2014 (control) | 99 | 2005–2006 | All co-infected patients >15 years newly diagnosed with TB or HIV | Guatemala | TB Hospital | Quasi-experimental (historical control) | Extensive HCW training in HIV/TB co-infection (40% of providers received additional training in HIV integrated care through national 8-month diploma program); systematic and provider-level co-located HIV testing; facility-level co-located ART initiation **Co-location (F–ST and Tx; P–ST only)** + HCW Train, Syst HIV T | 11.92 (5.46, 26.05) | Serious |
|  | Ikeda, 2014 (Intervention) | 155 | 2008–2009 |  |  |  |  |  |  |  |
| [36] | Kaplan, 2016 (control) | 3749 | staggered | All newly registered patients with drug-susceptible TB at participating facilities | South Africa | TB Clinics (within PHFs) | Quasi-experimental (historical control) | In-clinic TB educational sessions for all TB patients and HIV educational sessions for HIV-positive TB patients (patient education) performed by dedicated staff (adherence counsellors/ lay HCWs), following HCW training **Educ/Couns + Dedic Person** + HCW Train | 1.10 (1.07, 1.14) | Moderate |
|  | Kaplan, 2016 (Intervention) | 3411 | staggered |  |  |  |  |  |  |  |
| [37] | Kerschberger, 2012 (control) | 100 | 2008 | All TB/HIV patients >16 years not yet on ART and registered for TB treatment at participating facilities | South Africa | PHF | Quasi-experimental (historical control) | Systematic and provider-level co-located HIV testing (for PWTB); minor operational improvements including combined health information system, patient filing system (with medical notes, screening tools, prescription charts) and monitoring/evaluation; provider-level co-located ART initiation; oversight of integrated program by dedicated personnel (facility manager) **Co-location (F, P–ST and Tx)** + HCW Train, Oper Improv, Syst HIV T | 1.6 (1.11, 2.29)[d] | Moderate |
|  | Kerschberger, 2012 (Intervention) | 88 | 2009 |  |  |  |  |  |  |  |
| [24] | Kufa, 2017 (control) | 160 | 2011–2014 | All patients >18 years newly diagnosed with TB, HIV, or both | South Africa | PHFs | Cluster-randomized controlled trial | Task-shifting of TB screening from nurses to lay workers (Screening Officers); addition of dedicated personnel (Integration Officers) to support delivery of previous efforts towards TB/HIV collaboration **Dedic Person** +Task Shift | 0.99 (0.64, 1.54)[e] | RCT—HIGH |
|  | Kufa, 2017 (Intervention) | 224 |  |  |  |  |  |  |  |  |
| [38] | Louwagie, 2012 (control) | 233 | 2008–2009 | All TB patients newly diagnosed with HIV at participating facilities | South Africa | Hospitals/ PHFs | Quasi-experimental (historical control) | Facility- level co-location of ART initiation. **Co-location (F–Tx only)** | 1.58 (1.31, 1.91) | Serious |
|  | Louwagie, 2012 (Intervention) | 105 |  |  |  |  |  |  |  |  |

(Continued)

**Table 2.** (Continued)

**PICO 1: HIV TESTING (population = persons with active TB)**

| Ref | Author, year (arm) | Sample size | Study period (years) | Study Population | Country | Setting | Study Design | Intervention (summary) | Outcome RR (95% CI) | Risk of Bias |
|---|---|---|---|---|---|---|---|---|---|---|
| [28] | Mwinga, 2008 (control) | 196 | 2004–2005 | All TB patients registered at a participating facility | Zambia | Hospitals/ clinics | Non-randomized, cluster-controlled | Control: Referral (no co-location) between HTB/ HIV services. Int 2: Systematic and provider-level co-located HIV testing (for PWTB); HCW training to deliver as part of routine clinical care. **Co-location (F, P–ST only)** + HCW Train, Syst HIV T | 0.53 (0.45, 0.63) | Serious |
| | Mwinga, 2008 (Intervention 2) | 600 | 2006 | | | | | | | |
| [39] | Ogarkov, 2016 (control) | 84 | 2014 | All TB patients >15 years (excluding prisoners) newly diagnosed with HIV at participating hospital | Russia | TB Hospital | Quasi-experimental (historical control) | Major operational improvements including expedition of CD4 cell count and viral load testing + administrative prioritization of ART requests for co-infected patients through weekly cohort reviews of all PLHIV; patient education tailored to people with HIV and TB **Educ/Couns** + Oper Improv | 3.22 (1.92, 5.41) | Serious |
| | Ogarkov, 2016 (Intervention) | 82 | 2015 | | | | | | | |
| [40] | Owiti, 2015 (control) | 458 | 2010–2012 | All TB patients not yet on ARTs registering participating facilities | Kenya | Hospitals/ PHFs | Non-randomized, cluster-controlled | Control: Referral (no co-location) Int 1: Facility-level co-located testing and treatment Int 2: Facility-level co-located testing, provider-level co-located treatment Int 3: Provider-level co-located testing and treatment All interventions: onsite training and mentorship of HCW training, appointment of TB-HIV focal point person to oversee activities, patient education, major operational improvements including improved filing and record keeping, synchronised follow-up, key infection control practices and **Co-location (F, P–ST and Tx)** + Educ/Couns, Dedic Person + HCW Train | Arm 1: 1.32 (0.95, 1.83) Arm 2: 1.69 (1.42, 2.02) Arm 3: 1.53 (1.29, 1.81) | Moderate |
| | Owiti, 2015 (Intervention 1) | 39 | | | | | | | | |
| | Owiti, 2015 (Intervention 2) | 117 | | | | | | | | |
| | Owiti, 2015 (Intervention 3) | 167 | | | | | | | | |
| [41] | Van Rie, 2014 (control) | 373 | 2010–2012 | All patients >18 years diagnosed with TB and HIV, not yet on ARTs at a participating clinic | DRC | PHFs | Quasi-experimental (historical control) | Task-shifting of CD4-stratified ART initiation from clinicians to TB nurses; provider-level co-location of ART initiation. **Co-location (P–Tx only)** + Task Shift | 4.15 (3.28, 5.25) | Moderate |
| | Van Rie, 2014 (Intervention) | 513 | | | | | | | | |

**PICO 2: TB CASE-DETECTION (population = persons with known HIV)**

| Ref | Author, year (arm) | Sample size | Study period (years) | Study Population | Country | Setting | Study Design | Intervention (summary) | Outcome RR (95% CI) | Risk of Bias |
|---|---|---|---|---|---|---|---|---|---|---|
| [25] | Agarwal, 2018 (control) | 380 | 2012–2015 | All patients >18 years attending AIDS/TB facilities offering inpatient intensive treatment within 6 distinct oblasts | Ukraine | TB clinics/ HIV clinics (standalone) | Non-randomized, cluster- controlled before-and-after study | Facility-level co-located and systematic HIV testing (for PWTB) and systematic TB screening for (PLHIV); HCW training in caring for co-infected patients; major operational improvements including development of electronic data-management system and other capacity building initiatives to institutionalize best practices in TB-HIV care **Co-location (F–ST only)** + HCW Train, Oper Improv, Syst HIV T, Syst TB ST | 1.56 (1.08, 2.25) | Moderate |
| | Agarwal, 2018 (Intervention) | 402 | | | | | | | | |

(Continued)

**Table 2.** (Continued)

**PICO 1: HIV TESTING (population = persons with active TB)**

| Ref | Author, year (arm) | Sample size | Study period (years) | Study Population | Country | Setting | Study Design | Intervention (summary) | Outcome RR (95% CI) | Risk of Bias |
|---|---|---|---|---|---|---|---|---|---|---|
| [23] | Auld, 2020 (control) | 8622 | 2010–2012 | All new HIV clinic attendees > 12 years (excluding newly prison population) who started ART at or after study enrollment | Botswana | HIV clinic (within hospital/ PHF) | Stepped-wedge cluster randomized trial | Int 1: Systematic TB screening for PLHIV at all visits ("intensified case finding"); HCW training (clinic and lab personnel); support from dedicated personnel (additional nurses); minor operational improvements including checklists/ job aids to standardize implementation, and regular supervisory visits. Int 2: Int 1 + sputum smear microscopy replaced with Gene Xpert. **Dedic Person** + HCW Train, Oper Improv, Syst TB ST | Arm 1 vs. 2: 3.33 (2.55,4.36) Arm 1 vs. 3: 1.20 (0.94,1.52) | Moderate |
| | Auld, 2020 (Intervention 1) | 4093 | 2012–2013 | | | | | | | |
| | Auld, 2020 (Intervention 2) | 1724 | 2012–2014 | | | | | | | |
| [42] | Hermans, S 2012 (control) | 9931 | 2010 | All adult (age not specified) patients attending the clinic who were not already diagnosed or on TB treatment | Uganda | HIV clinic (standalone) | Quasi-experimental (historical control) | Twice daily patient education presentations on TB and TB-HIV co-infection and the ICF screening questions, encouraging patients to self-identify if they had any of the described symptoms (cough >2 weeks, hemoptysis, fever>3 weeks, LOW >3kg/month); delivered in HIV clinic waiting area by two trained peer supporters. **Educ/Couns + Peer Supp** | 1.22 (0.98,1.52) | Serious |
| | Hermans, S 2012 (Intervention) | 10525 | 2010 | | | | | | | |
| [43] | Kanara, 2008 (control) | 1228 | 2003–2005 | All PLHIV or PWTB attending a participating facility | Cambodia | TB clinics/ HIV clinics (standalone) | Quasi-experimental (historical control) | Monthly educational meetings for TB/HIV staff (HCW training); minor operational improvements including supplemental data collection form to collect information about HIV status, referral for HIV testing, CPT status and AIDS care status for all TB patients; systematic patient education on risk of TB among all PLHIV **Educ/Couns** + HCW Train, Oper Improv | 1.53 (1.18,1.98) | Serious |
| | Kanara, 2008 (Intervention) | 751 | 2005 | | | | | | | |
| [44] | Mathebula, 2020 (control) | 870 | 2012–2013 | All new HIV clinic attendees > 12 years who screened positive for TB | Botswana | HIV clinics (standalone) | Quasi-experimental (historical control) | HCW training and onsite mentorship to improve sputum induction and nebulization techniques, infection control; patient education and assistance for sputum induction; minor operational improvements including sputum collection job aid, tracking log sheet and regular monitoring by nurse supervisors to evaluate quality of screening/ documentation. **Educ/Couns** + HCW Train, Oper Improv | 1.24 (0.96–1.63) | Low |
| | Mathebula, 2020 (Intervention) | 993 | 2013–2014 | | | | | | | |
| **PICO 2: ATT INITIATION (population = persons with known HIV and active TB)** | | | | | | | | | | |
| Ref | Author, year (arm) | Sample size | Study period (years) | Study Population | Country | Setting | Study Design | Intervention (summary) | Outcome RR (95% CI) | Risk of Bias |
| [25] | Agarwal, 2018 (control) | 297 | 2012–2015 | All patients >18 years attending AIDS/TB facilities offering inpatient intensive treatment within 6 distinct oblasts | Ukraine | TB clinics/ HIV clinics (standalone) | Non-randomized, cluster- controlled before-and-after study | Facility-level co-located and systematic HIV testing (for PWTB) and systematic TB screening for (PLHIV); HCW training in caring for co-infected patients; major operational improvements including development of electronic data-management system and other capacity building initiatives to institutionalize best practices in TB-HIV care **Co-location (F–ST only)** + HCW Train, Oper Improv, Syst HIV T, Syst TB ST | 0.99 (0.99,1.00) | Moderate |
| | Agarwal, 2018 (Intervention) | 565 | | | | | | | | |

(*Continued*)

**Table 2.** (Continued)

| | | | | | HIV clinic (standalone) | Quasi-experimental (historical control) | | |
|---|---|---|---|---|---|---|---|---|
| **PICO 1: HIV TESTING (population = persons with active TB)** | | | | | | | | |
| [42] | Hermans, S 2012 (control) | 9931 | 2010 | All adult (age not specified) patients attending the clinic who were not already diagnosed or on TB treatment | Uganda | | | Serious |
| | Hermans, S 2012 (Intervention) | 10525 | 2010 | | | | Twice daily patient education presentations on TB and TB-HIV co-infection and the ICF screening questions, encouraging patients to self-identify if they had any of the described symptoms (cough >2 weeks, hemoptysis, fever>3 weeks, LOW >3kg/month); delivered in HIV clinic waiting area by two trained peer supporters. **Educ/Couns + Peer Supp** | 0.97 (0.92, 1.03) |

ART, antiretroviral therapy; CPT, Co-trimoxazole Preventive Therapy; Co-location types: F, same facility; P, same provider, ST, TB screening and testing or HIV testing; Tx, TB or HIV treatment; Dedic Person, dedicated personnel; Educ/Coun, patient education/counselling; Financ Supp, patient financial support; HCW train, healthcare worker training; Oper Improv, operational improvements; PHF, primary healthcare facility; Peer Supp, patient peer support; Syst HIV T, systematic HIV testing; Syst TB ST, systematic TB screening and testing; Task Shift, task-shifting.

[a] Only first authors are listed.

[b] Interventions are summarized and abbreviated. Bolded text represents the interventions analyzed for this review (some strategies were implemented as co-interventions but are now considered standard of care and hence not analyzed).

[c] HR adjusted for difference-in-differences, as reported by authors.

[d] HR adjusted for age, gender, CD4 count, previous TB treatment initiation, as reported for by authors.

[e] RR adjusted for randomization strata, sex, age group, country of birth, education level, marital status, employment status, SEP level, CPT at enrolment, as reported by authors.

**Table 3. Definitions of intervention analyzed.**

| Intervention | Abbreviation | N [a] | Definition |
|---|---|---|---|
| Co-location | Co-location | 18 | HIV and TB care were co-located, based on<br>1) Type of service co-located:<br>ST: screening and/or testing (including sputum collection for TB)<br>Tx: treatment<br>2) Level of co-location<br>F: facility, services delivered at the same by different provider/s<br>P: provider, services delivered at the same facility and by same provider/s |
| Patient education/ counselling | Educ/Coun | 6 [b] | Patients received education and/or counselling via one to one or group sessions on diverse topics (e.g., TB, HIV, TB-HIV, ART, sputum induction methods) that went beyond standard-of-care (e.g., pre and post HIV testing counselling). |
| Dedicated personnel | Dedic Person | 5 [b] | Personnel (other than patient peers) were introduced to support diverse TB-HIV related activities (e.g., TB screening/ testing, HIV testing, treatment monitoring, case management, HCW supervision, clinic or regional program coordination). |
| Patient peer support | Peer Supp | 3 [b] | Patient peers (i.e., PLHIV, people with past TB) were used to support diverse TB-HIV related activities (e.g., assist with operational changes, deliver patient education) |
| Patient financial support | Financ Supp | 1 [b] | Patients attended workshops on income-generation through microenterprise, microcredits, and vocational training plus poverty reduction techniques including food and cash transfers. |

[a] N = study intervention arms. (Reflects interventions that were present in the intervention arm only; some studies included an intervention in standard of care and intervention arms but were focused on comparing the effect of another intervention).

[b] Excludes study arms that also had co-location as an intervention.

and TB at the same facility and by the same provider (Fig 2). In all eight studies reporting on HIV testing, test rates improved when co-located with TB services, regardless of whether it was at the same facility or by the same provider, and there was no apparent difference between the two models of increasing co-location. Likewise, ART initiation improved with almost all combinations of co-location, except for two studies where testing was co-located at the same facility and treatment delivered by the same provider [33,40], and one study where testing alone was delivered by the same provider [28]. In two of these three studies, negative or null effects on ART initiation rates were attributed to clinics' inability to absorb increased numbers of PLHIV identified through the co-located model [28,33]. There was no strong evidence to

**Table 4. Definitions of standard of care strategies implemented as co-interventions.**

| Standard of care strategy [a] | Abbreviation | N [b] | Definition |
|---|---|---|---|
| Health care worker (HCW) training | HCW Train | 19 | Existing HCW underwent training in TB-HIV care for diverse periods (e.g., single workshop, long courses) on diverse topics (e.g. general TB-HIV care, universal screening/testing for TB and/or HIV in people with one known infection, referral to dual/follow-up care, and specific issues such as sputum induction techniques for TB testing, guidance on co-treatment and infection control methods). |
| Task-shifting | Task Shift | 2 | Tasks relevant to TB-HIV care (e.g., ART initiation/monitoring, TB screening) were shifted from diverse specialized HCW to less specialized workers (e.g., clinicians to nurses, or nurses to lay counsellors). |
| Systematic HIV testing | Syst HIV T | 7 | HIV testing was systematized for all patients with known active TB through an opt-out approach. Testing was provider-initiated. |
| Systematic TB screening | Syst TB ST | 3 | TB screening was systematized for all patients with known HIV infection through use of a new standardized screening tool (e.g., form or algorithm based on WHO guidance). Screening was provider-initiated. |
| Operational improvements | Oper Improv | 15 | Improvements were made to facilities to support processes of TB-HIV service integration. Improvements ranged from minor (e.g., record-keeping via use of forms/logs, checklists/job aids, staff meetings, or HCW mentorship or supervision) to major (e.g., dedicated TB-HIV clinic days, fast-tracking services for coinfected patients, development of electronic data-management system, or multiple minor improvement/s). |

[a] Reflects strategies that are now considered standard of care, present in the intervention arm only; some studies included such strategies in both arms but were focused on studying the effect of another intervention.

[b] N = study intervention arms.

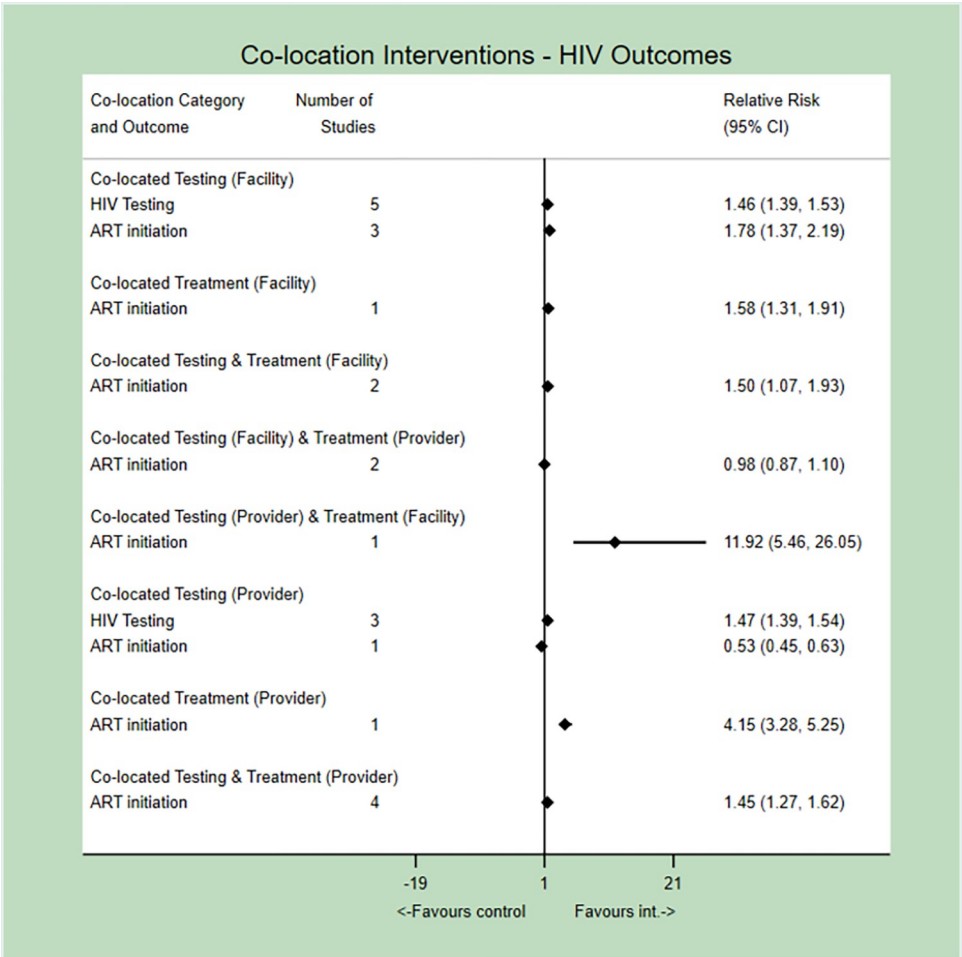

**Fig 2. Meta-analysis and forest plot of the effect of co-location interventions (at the facility vs. within the same provider; for just testing, treatment initiation, or both) on outcomes of HIV testing and ART initiation for people with TB (PICO 1).**

suggest that a single model of co-location out-performed others in improving outcomes in PICO 1.

For PICO 2, we identified one combination of co-location in a single study where TB screening and testing was co-located with HIV services [25] (Fig 3). This model improved TB case-detection, identifying an approximate additional 59 cases per 1,000 (RR 1.56, 95% 1.08–2.25). ATT initiation did not improve; the baseline rate was already 100%.

Five studies described needs and considerations pertinent to the success of co-location interventions [28,33,35,37,40]. They include dedicated counselling spaces, infection control measures (e.g., ventilation, UV lighting, outdoor clinics and sputum booths, personal protective equipment, and infection control officers), improved filing, records and communication, synchronized appointments including pharmacy services, and personnel for allied patient support such as nutrition, outreach, tracing and social/adherence support.

## Patient education and counselling

Patient education and counselling interventions covered a range of topics. For PICO 1, one study reported on HIV testing and two other studies on ART initiation; all demonstrated

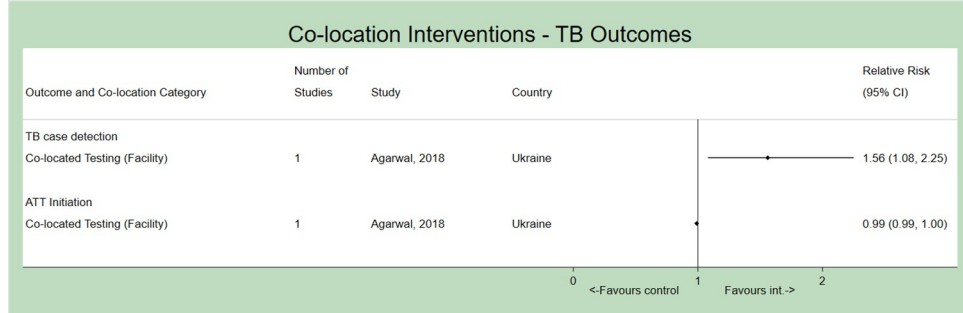

**Fig 3. Forest plot (not pooled) depicting the effect of co-location interventions (only observed at the level of the facility) on outcomes of TB diagnosis and treatment initiation for people living with HIV (PICO 2).** ATT = Anti-tuberculosis Treatment.

significant improvements (Fig 4). Psychosocial counselling about depression and substance use, among other topics, to households affected by HIV (and TB) improved HIV test rates in a community-based study in Peru (RR 3.17, 95% CI 2.24–4.49) [22]. Educating and counselling about TB-HIV and ART improved ART initiation among TB patients at a hospital in Russia (RR 3.22, 95%CI 1.92–5.41) [39] and at clinics in South Africa (RR 1.10, 95% CI 1.07–1.14) [36]. Three studies implemented co-interventions (i.e., financial support and peer supporters or other dedicated personnel to deliver the educational component) [22,36,42]. Remaining studies implemented facility-specific operational improvements as well (e.g., supplemental data collection forms, expedited ART initiation for TB patients diagnosed with HIV) [39,43,44].

Patient education interventions were more heterogenous in studies examining PICO 2, ranging from group presentations to one-on-one education sessions delivered by diverse personnel including peer supporters, counsellors and nurses (Fig 4). All studies implemented co-interventions. TB case-detection improved in only one study in Cambodia where newly

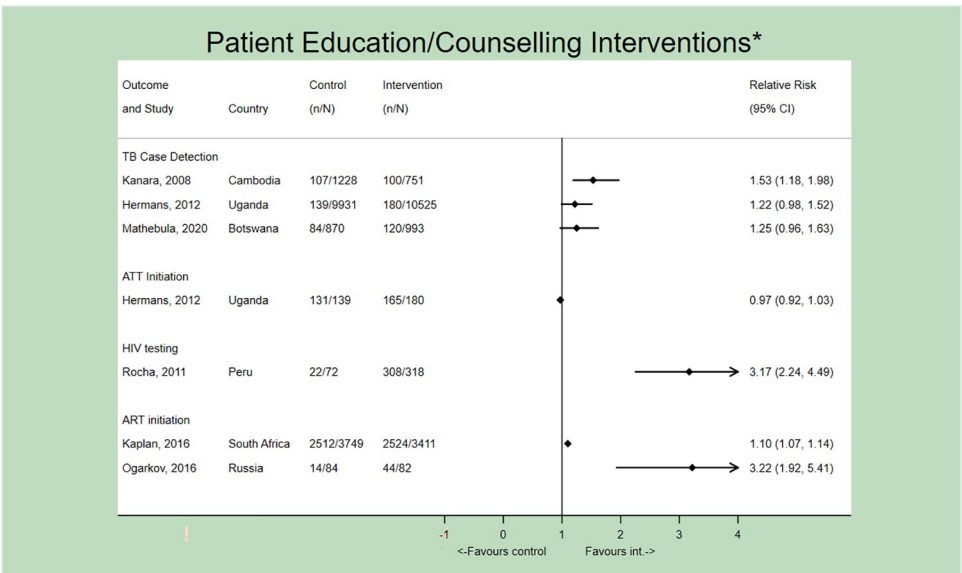

**Fig 4. Forest plot (not pooled) depicting the effect of patient education and counselling interventions on all outcomes.** *All studies implemented concurrent interventions and/or SOC strategies. Only the first author of each study is listed. ATT = Anti-tuberculosis treatment.

diagnosed PLHIV were educated about the increased risk of TB by HIV counsellors using a scripted message (RR = 1.53, 95% 1.18–1.98); health care worker training and facility-specific operational improvements were part of the intervention package [43]. In two other studies implemented at HIV clinics, one in Uganda where group presentations about TB were delivered by peer supporters [42], and one in Botswana where nurses educated patients on sputum induction techniques [44], TB case-detection did not improve. ATT initiation also did not improve in the one study reporting on this outcome; the baseline rate was high (94%) [42].

## Dedicated personnel

Dedicated personnel were introduced to support a variety of TB-HIV services. For PICO 1, having a dedicated HIV testing services counsellor systematically test all TB patients significantly improved the HIV test rate in a study in Nigeria (RR 2.73, 95% CI 2.33–3.22); importantly, systematic HIV testing for TB patients was absent at baseline [27]. Dedicated personnel did not however appear to have a strong effect on ART initiation in this and two other studies from South Africa, one utilizing professional nurses to support facility-level operations [24] and one utilizing adherence counsellors and lay workers to deliver patient education and counselling [36] (Fig 5). One possible reason for the lack of demonstrable effect on ART initiation with dedicated personnel may be a result of their involvement on earlier parts of the care cascade (i.e., conducting screening or testing, and not involved in linkage to treatment). These studies also implemented co-interventions, including co-location of TB and HIV services [24,27] and patient education and counselling [36].

For PICO 2, addition of dedicated nurses to support TB screening and testing improved TB case-detection in two arms of the same intervention at an HIV clinic in Botswana; however, this intervention also newly introduced systematic TB screening which was not present at baseline. [23] (Fig 5). No studies examined the effect of dedicated personnel on ATT initiation.

Two studies described considerations pertinent to the success of such interventions [28,36]. One study reported that provision of a transport allowance enabled personnel to attend the clinic [28], and another study recommended adopting a task-sharing approach to decongest clinics and facilitate an overall decreased consumption of clinic resources [36].

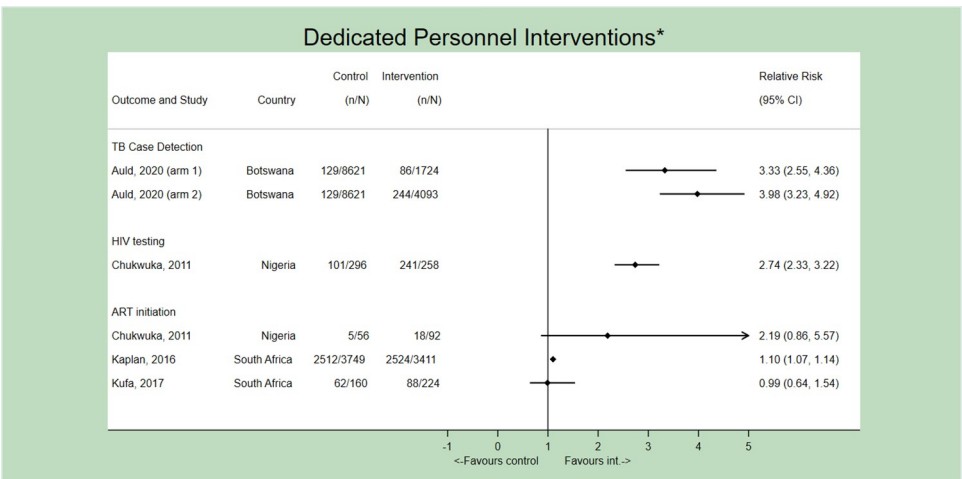

**Fig 5. Forest plot (not pooled) depicting the effect of dedicated personnel interventions on all outcomes.** *All studies implemented concurrent interventions and/or SOC strategies. Only the first author of each study is listed.

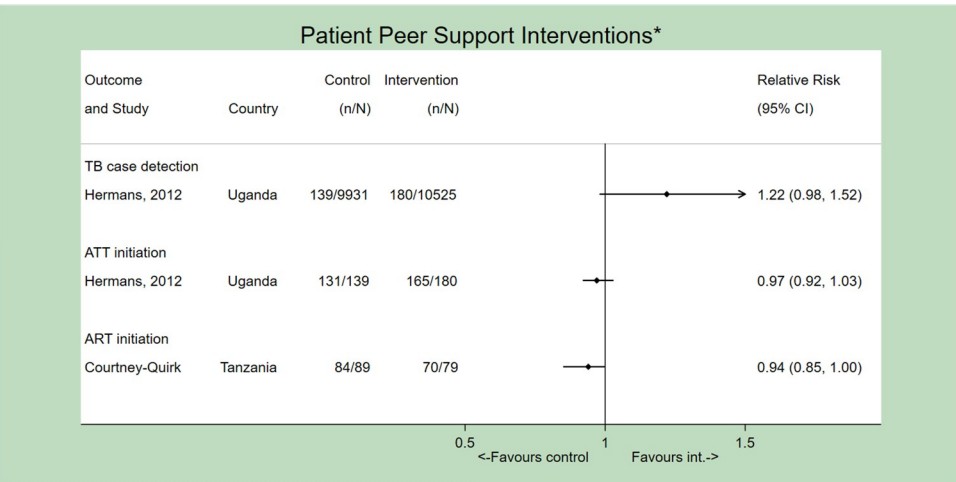

**Fig 6. Forest plot (not pooled) depicting the effect of patient peer support interventions on all outcomes.** *All studies implemented concurrent interventions and/or SOC strategies. Only the first author of each study is listed. ATT = Anti-tuberculosis treatment.

## Patient peer support

Patient peers were introduced to support services related to TB-HIV care. For PICO 1, only one study, from Tanzania, examined the effect of patient peer support on ART initiation and did not have an apparent effect. The rate of ART initiation was high at baseline (94.4%) [31] (Fig 6).

Likewise, for PICO 2, only one study from Uganda examined the effect of patient peer support on both TB case-detection and ATT initiation, with no apparent effect on either outcome [42] (Fig 6). Here, peer supporters gave group presentations about TB in HIV clinic waiting areas, encouraging patients to self-report TB symptoms. Authors believed this was ineffective due to stigma attached to self-identifying TB symptoms, language barriers, and potentially sub-optimal screening by peer supporters. The intervention was not designed to focus on ATT initiation, as peer support was delivered during an early part of the care cascade, during TB testing but before diagnosis. The baseline ATT initiation rate was also high (94.7%) [42].

Studies implementing peer support deemed the skills, fluency in local languages, and enthusiasm of peers to be key considerations [23,42,43].

## Patient financial support

For PICO 1, only one study examined the effect of patient financial support and demonstrated improved rates of HIV testing (RR = 3.17, 95% 2.24–4.49) [22] (Fig 7). Here, TB-affected households in Peru received food/cash transfers, vocational training, and microfinance strategies, in addition to psychosocial counselling. No studies examined the effect of patient financial support on PICO 2.

## General considerations for implementing TB-HIV interventions

Several studies described implementation considerations for specific types of interventions. Fourteen studies described general considerations for interventions seeking to link patients to TB and HIV services, which centered on facilities' and programs' capacity to absorb increases in new TB and/or HIV diagnoses (laboratory capacity and supply chain management for tests and treatment [25,26,30,32,37,42,44]), data management (creation of single patient files and

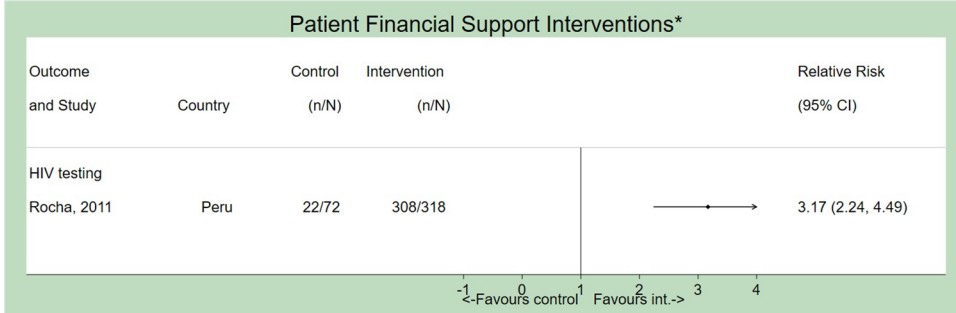

**Fig 7. Forest plot (not pooled) depicting the effect of patient financial support interventions on all outcomes.** [*] All studies implemented concurrent interventions and/or SOC strategies. Only the first author of each study is listed. ATT = Anti-tuberculosis treatment.

data sharing between TB and HIV programs [25,26,31,43]), human resources (health care worker burden, turnover and shortage [23,24,32,37,40,42]), health care worker competencies and perceptions (TB and/or HIV risk misperceptions, reluctance to manage TB and HIV, difficulties diagnosing TB in PLHIV or managing complications such as IRIS, and negative attitudes towards patients [25,32,37,40,43]); and patient perceptions (risk misperceptions, TB and/or HIV stigma, concerns about treatment side effects and clinic commutes to access treatment [25,42,43]). For a summary of all implementation barriers and facilitators by outcome, including cost analyses, see S8 File: Implementation Considerations.

## Discussion

This systematic review is the first instance of presenting the full spectrum of interventions seeking to improve identification of HIV or TB disease among people with one known infection, and initiation of co-treatment in people with both. We identified a several interventions supporting these points of linkage to the TB-HIV care cascades, which build upon the findings of other reviews that have focused on other points of linkage such as TB prevention in PLHIV or retention in HIV or TB care [6–11]. The results were presented to the WHO Service Delivery guidelines meeting in October 2020 to inform the 2021 Updated Recommendations on Service Delivery for Treatment and Care of People Living with HIV [45].

The most noteworthy findings of the review centered on the consistent effectiveness of co-located HIV and TB services in improving rates of HIV testing and ART initiation among people with TB, and likewise improving TB case-detection among PLHIV. This validates prior recommendations to link patients to integrated TB-HIV care by providing services at the same time and location [4]. Several studies reported on the time saved and coordination gained from attending to both infections within a single facility as compared to referring patients to another site [29,34,38]. Nonetheless, in 2019, of the 30 high burden TB-HIV countries only 11 had reported having countrywide co-location of HIV and TB testing, and only five countries reported delivering ART and ATT within the same facility [5]. The studies described herein offer blueprints to guide the future operationalization of co-location interventions and help fill these gaps.

Also of note, several studies suggested that providing HIV and TB services at not only the same facility but also by the same provider could help to reduce patient "juggling" and losses to follow-up seen when services were delivered by different providers or clinics within a single facility [40]; typically, due to perceived patient stigmas [25,32,42] or the poor integration of medical information systems [37]. However, heightening the degree of integration to the level

of the same provider (or same set of providers) did not have a consistently greater effect on HIV testing or ART initiation when compared to facility-level integration.

Overall, most interventions reviewed were multi-component and included facility-specific operational improvements as well as strategies that are today considered to be standard of care. Nonetheless, when implemented as part of a broader intervention package, the introduction of dedicated personnel to support delivery of TB-HIV services improved HIV testing in people with TB and TB case-detection rates in PLHIV. Similarly, patient education and counselling about TB-HIV coinfection and financial support improved HIV testing and ART initiation in people with TB [22,32,36]. The effect on TB case-detection in PLHIV was mixed but rates of TB testing universally improved in studies implementing patient education and counselling as well as peer support interventions, pointing to critical achievements in the provision and uptake of TB test services [42–44].

Technical or resource barriers reported within reviewed studies, may have contributed to the observed lack of effect or consistent effect of interventions that held promise. Many of these barriers have been previously described [5,46–50], and point to a need for wider health systems strengthening to build human resource competencies, and programmatic foresight to allocate resources to accommodate potential increases in patient volume. Concurrent efforts are also needed to correct prevailing misperceptions and gaps in knowledge around TB-HIV risk, dual diagnosis, and co-treatment. Such investments will likely support continuity of care for patients with complex multiple conditions in the face of disruptions to HIV and TB programs such as those experienced during the COVID-19 pandemic [51,52].

This review also highlighted gaps in innovation with respect to the published literature. For example, we found relatively few studies (5/23) seeking to improve TB testing and treatment among PLHIV, and no studies implemented in settings with low baseline rates of TB treatment initiation. Community-based interventions, where the most vulnerable populations may be more easily reached, and that have been shown to improve TB screening and testing [53] as well as HIV testing [54] in the general population, were scarce. Other novel intervention designs that have improved outcomes in HIV or TB, such as differentiated service delivery and m-Health or virtual interventions [55–58], were also not identified. Programs may consider adapting successful models to better link people with HIV and TB disease to integrated care. Not surprisingly, given the difficulties of designing RCTs around various models of integrated TB-HIV care, most evidence was sourced from observational studies.

The review has several limitations. First, only interventions implemented among participants with known HIV or TB disease were included. Studies were thus largely facility based. Studies assessing linkages between HIV and TB services from community to facility among people not yet diagnosed with either infection, or studies which sought to integrate TB screening and HIV testing for the broader population were also excluded, though they have reported improved identification of PLHIV and TB disease [59–61]. Second, interventions and standards of care were highly heterogenous with much overlap, particularly co-location and provider-initiated HIV testing. To mitigate this limitation, we only pooled results for co-location interventions (by level of co-location), which we deemed methodologically homogenous; for the remaining interventions of interest, we narratively described and compared interventions and their effects. Studies also spanned wide timelines, before and after important policy changes were instituted, such as opt-out HIV testing in 2007 [17] and ART initiation for all people with TB regardless of CD4 count in 2016 [18] amidst diverse country and population contexts. Third, few studies reported on intervention cost or feasibility, and thus successes may be limited to relatively better resourced environments within LMICs. Finally, we included studies indexed in only two databases (Medline and EMBASE), and only in English, therefore potentially limiting the thoroughness of our results.

## Conclusion

In supporting the latest WHO recommendations on HIV service delivery, this review emphasizes the effectiveness of co-locating HIV and TB testing and treatment services to improve outcomes in HIV testing, ART initiation and TB case-detection in people with HIV and TB disease. Various models of HIV and TB test and treatment service co-location are exemplified in the studies reviewed that offer critical insights into implementation facilitators and barriers. The evidence further suggests that provision of joint services at the same facility, even if delivered by distinct sets of providers, may be sufficient to achieve improvements in early TB-HIV outcomes. Other patient-centered interventions such as financial or peer support and allocating dedicated personnel for TB-HIV service delivery show promise. Future implementation research would benefit from evaluating the distinct effectiveness of these patient-centered interventions, and of adapting community-based approaches, virtual approaches, and differentiated service delivery models to address patient and health system needs in the context of TB-HIV co-morbidity.

## Supporting information

**S1 File. PRISMA checklist.**
(DOCX)

**S2 File. Systematic Review Protocol (PROSPERO).**
(PDF)

**S3 File. Search strategy.**
(DOCX)

**S4 File. Quality assessments.**
(DOCX)

**S5 File. Grade evidence profiles.**
(DOCX)

**S6 File. Secondary outcomes.**
(DOCX)

**S7 File. Interventions identified in control and intervention arms.**
(DOCX)

**S8 File. Implementation considerations.**
(DOCX)

## Acknowledgments

We thank McGill University librarians Genevieve Gore and Martin Morris for assisting with our search strategy, and authors of primary papers contacted in the process of our review.

## Author Contributions

**Conceptualization:** Cheryl Johnson, Annabel Baddeley, Satvinder Singh, Amrita Daftary.

**Data curation:** Angela Salomon, Stephanie Law.

**Formal analysis:** Angela Salomon, Stephanie Law, Satvinder Singh, Amrita Daftary.

**Funding acquisition:** Satvinder Singh, Amrita Daftary.

**Methodology:** Angela Salomon, Satvinder Singh.

**Supervision:** Amrita Daftary.

**Validation:** Stephanie Law.

**Visualization:** Amrita Daftary.

**Writing – original draft:** Angela Salomon, Amrita Daftary.

**Writing – review & editing:** Angela Salomon, Stephanie Law, Cheryl Johnson, Annabel Baddeley, Ajay Rangaraj, Satvinder Singh, Amrita Daftary.

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
