## [Decision Letter · Decision Letter 0]

14 Feb 2022

PONE-D-21-39116Interventions to improve linkage along the HIV-tuberculosis care cascades in low- and middle-income countries: a systematic reviewPLOS ONE

Dear Dr. Daftary,

Thank you for submitting your manuscript to PLOS ONE. After careful consideration, we feel that it has merit but does not fully meet PLOS ONE’s publication criteria as it currently stands. Therefore, we invite you to submit a revised version of the manuscript that addresses the points raised during the review process.

Based on my own assessment and that of reviewers the manuscript have merit. However, it requires further revision before it can be considered for publication.Please ensure that your decision is justified on PLOS ONE’s publication criteria and not, for example, on novelty or perceived impact.

We look forward to receiving your revised manuscript.

Kind regards,

Gabriel O Dida, PhD

Academic Editor

PLOS ONE

Journal Requirements:

“This study was funded by United States Agency for International Development through the World Health Organization. The funders had no role in the design, data collection, analysis and decision to publish the results. The corresponding author (AD) had full access to all the data in the study and had final responsibility for the decision to submit for publication.”

Please note that funding information should not appear in other areas of your manuscript. We will only publish funding information present in the Funding Statement section of the online submission form.

“This study was funded by United States Agency for International Development through the World Health Organization. The funders had no role in study design, data collection and analysis, decision to publish, or preparation of the manuscript. The corresponding author (AD) had full access to all the data in the study and had final responsibility for the decision to submit for publication.”

“This study was funded by United States Agency for International Development through the World Health Organization. The funders had no role in study design, data collection and analysis, decision to publish, or preparation of the manuscript. The corresponding author (AD) had full access to all the data in the study and had final responsibility for the decision to submit for publication.”

“This study was funded by United States Agency for International Development through the World Health Organization. The funders had no role in study design, data collection and analysis, decision to publish, or preparation of the manuscript. The corresponding author (AD) had full access to all the data in the study and had final responsibility for the decision to submit for publication.”

Reviewers' comments:

Reviewer's Responses to Questions

**Comments to the Author**

1. Is the manuscript technically sound, and do the data support the conclusions?

Reviewer #1: Yes

Reviewer #2: Yes

Reviewer #3: Yes

Reviewer #4: Yes

Reviewer #5: Yes

2. Has the statistical analysis been performed appropriately and rigorously? 

Reviewer #1: Yes

Reviewer #2: Yes

Reviewer #3: N/A

Reviewer #4: I Don't Know

Reviewer #5: Yes

3. Have the authors made all data underlying the findings in their manuscript fully available?

Reviewer #1: Yes

Reviewer #2: Yes

Reviewer #3: Yes

Reviewer #4: Yes

Reviewer #5: Yes

4. Is the manuscript presented in an intelligible fashion and written in standard English?

Reviewer #1: Yes

Reviewer #2: Yes

Reviewer #3: Yes

Reviewer #4: Yes

Reviewer #5: Yes

5. Review Comments to the Author

Reviewer #1: Dear Authors,

Overall, the manuscript is interesting and have much contribution for educational and clinical field. However, I have some suggestions to consider by authors.

Abstract

- Line 2: as author mention on line 46. The study also performed meta-analysis. I suggest the title also mention “ A systematic review and metanalysis” .

- Line 40: Please add study aim.

- Line 43 to 45: move aiming …. Countries to introduction.

- In the methods, please add the keywords and or terms and Boolean operator used in the study.

- Line 44. Please add the databases used in the study.

- Line 59-60: “evidence….limited” please elaborate the sentence or give more explanation.

- Line 62: please provide the implication of the study.

- Please provide keywords after abstract.

Introduction

- The introduction is quietly concise. However, by adding some information related the prevalence of HIV-TB would be explaining to the readers about how important HIV-TB linkage is.

- AS you present the information in the paragraph 2, you stated the gaps. Please using firm statements and also provide the novelty of the study by compared by previous study. It would be presenting more information for readers.

- Anyway, do you used theoretical frameworks in this study? if so, please state it.

Methods

- Line 97: PRISMA and SWiM are stands from? Why do you use both PRIMS and SWiM? Please provide explanation. You may also state the study design properly.

- Line 114: Please add the information about Boolean operator used in the study. Besides, I have seen the supp file 2. Please make a separate information about searching strategy each database.

- Line 115: what do you mean “/” is it stands for “OR”? please clarify.

- On the PRISMA figure, please refer to PRISMA 2020 templates.

- For the quality assessment, how many authors contributed for it?

Results & Discussion

- Overall, the result & discussion are well explained.

Reviewer #2: 1- Do the authors have any particular reasons to use only Medline (OVID), Embase, Embase Classic? It is well known that PubMed is also a resourceful database including all the citations from MEDLINE plus NIH projects and PMC which also could be a good choice if one intends to search for literature related to TB and HIV.

2- An important keyword could be AIDS, which is missing. However, I believe the authors have covered that in their search strategy.

3- The objective is the linkage between TB and HIV care in LMICs countries; however, it seems that it has not been considered in search strategies and the authors search for all the available records. Moreover, how did the authors define the LMICs? Based which definition?

4- Line 134: The authors mentioned that the “discrepancies were resolved with a third reviewer (AD, VS)”. However, there are two names in the parenthesis. Please correct.

5- Why did the authors choose to use fixed-effect meta-analysis and not a random-effect model?

Reviewer #3: This systematic review is the first to assess the full spectrum of interventions to improve identification of HIV or TB disease among people with one known infection, and initiation of co-treatment in people with both. Based on the enormous individual and social burden of HIV, TB and HIV-TB co-infection, information of this nature is critically important.

The problem/ research question is well motivated and the argument for the study has been developed in a clear manner. The introduction highlights why the study is important and defines the purpose of the work and its significance. The methodology that has been applied, provides an evidence base for the themes that have been identified. Themes are discussed with insight into possible reasons for findings - in this regard, the paper makes a meaningful contribution to evaluating interventions that can address identification and initiation of co-treatment for HIV, TB and HIV-TB co-infection. Based on statistical heterogeneity (I squared), the finding that joint services at the same facility are likely to result in the best HIV and TB outcomes provide direction for future interventions. The narrative synthesis of facilitators and barriers adds additional value

The paper is well-written and comprehensive, but the abstract does not do justice to the full paper (e.g. indicate how homogeneity was defined; and refer to findings of meta-analysis). The abstract could benefit from language editing.

Reviewer #4: This is a well written manuscript. There are a few issues to revise if possible. Please justify why LMICs were reviewed and whether there were gold standards to compare with for each grouping (PICO1 and PICO2) eg from higher income countries etc. For Table 1, can the authors be described fully or indicate in the key below that you are only listing the first author. Additionally, please link the papers in Table 1 to the reference list to make it easier to read.

Reviewer #5: The authors have aggregated studies looking at HIV and TB co-morbidities and interventions to treat both diseases, and the nuances that come along with it, with the ultimate goal of improving patient care and treatment.

They approach the literature analysis very thoroughly, and clearly held their analyses of articles to a high standard for inclusion. Their workflow was very clear, and the supplemental data was very helpful for anyone wishing to replicate their workflow.

I detected essentially no grammatical errors, and the text was well organized and easy to read.

Honestly, a great paper, and a breath of fresh air after slogging through a bunch of terrible article reviews.

*My only issue is that your figure legends are missing. They just need a little extra info/description, especially Figure 2, which will set the precedence for each other figure's stats.

6. PLOS authors have the option to publish the peer review history of their article (what does this mean?). If published, this will include your full peer review and any attached files.

Reviewer #1: No

Reviewer #2: **Yes: **Omid Dadras

Reviewer #3: **Yes: **Corinna May Walsh

Reviewer #4: No

Reviewer #5: **Yes: **Jonathan LeCureux

---

## [Author Response · Author response to Decision Letter 0]

29 Mar 2022

Reviewer 1 

Overall, the manuscript is interesting and have much contribution for educational and clinical field. However, I have some suggestions to consider by authors.

Abstract

1. Line 2: as author mention on line 46. The study also performed meta-analysis. I suggest the title also mention “ A systematic review and metanalysis” .

Thank you. The title has been amended to: “Interventions to improve linkage along the HIV-tuberculosis care cascades in low- and middle-income countries: a systematic review and meta-analysis”.

2. Line 40: Please add study aim. 

The abstract introduction has been amended: “In support of global targets to end HIV/AIDS and tuberculosis (TB) by 2030, we reviewed interventions aiming to improve TB case-detection and anti-TB treatment among people living with HIV (PLHIV) and HIV testing and antiretroviral treatment initiation among people with TB disease in low- and middle-income countries (LMICs).”

3. Line 43 to 45: move aiming …. Countries to introduction. 

We have implemented this suggestion. 

4. In the methods, please add the keywords and or terms and Boolean operator used in the study. 

We have added this detail in the abstract: We performed random-effects effect meta-analyses (DerSimonian and Laird method) …” and included keywords after the abstract.

5. Line 44. Please add the databases used in the study.

We have added this detail in the abstract: "We conducted a systematic review of comparative (quasi-)experimental interventional studies published in Medline or EMBASE…” 

6. Line 59-60: “evidence….limited” please elaborate the sentence or give more explanation.

We have amended the abstract to reframe the point: “A majority … reported on HIV outcomes in people with TB (n=18).”

7. Line 62: please provide the implication of the study.

We have re-worded the abstract to highlight the implications more clearly: “This review provides operational clarity on intervention models that can support early linkages between the TB and HIV care cascades. The findings have supported the World Health Organization 2020 HIV Service Delivery Guidelines update. Further research is needed to evaluate the distinct effect of education and counselling, financial support, and dedicated personnel interventions, and to explore the role of community-based, virtual, and differentiated service delivery models in addressing TB-HIV co-morbidity.” 

8. Please provide keywords after abstract. 

We have included the following keywords: “TB-HIV coinfection, integrated care, HIV testing, TB case detection, ART initiation, TB treatment initiation, HIV, TB, AIDS, low and middle-income countries, co-location, interventions”.

Introduction

1. The introduction is quietly concise. However, by adding some information related the prevalence of HIV-TB would be explaining to the readers about how important HIV-TB linkage is.

We highlighted that TB is a leading opportunistic infection and cause of mortality for PLHIV, and that HIV accelerates TB development and progression. We have now also specified the two are linked: “through weakening of the immune system”. 

2. As you present the information in the paragraph 2, you stated the gaps. Please using firm statements and also provide the novelty of the study by compared by previous study. It would be presenting more information for readers.

This paragraph has been re-written to highlight gaps in the literature / TB-HIV discourse, and novelty of the review: “Gaps in linkage between the HIV and TB care cascades may be partly explained by the inadequate adoption and implementation of global recommendations within country programs, slow scale-up of new technologies, particularly rapid TB diagnostics, and disparate funding, monitoring and evaluation systems for HIV and TB. Programmatic guidance for the integration of TB and HIV services such as successful models and implementation considerations is also limited. Published reviews have focused on the effect of specific interventions such as patient food support, workplace programs, and private-public partnerships, and focused on prevention of TB disease in PLHIV, and adherence to ART and/or ATT in those receiving dual treatment. This systematic review uniquely assesses the full spectrum of non-clinical interventions targeted to patients, providers and programs in low- and middle-income countries (LMICs), and focusses on two critical underexplored outcomes in the TB-HIV care cascade – testing and diagnosis of TB or HIV co-morbidity in people with one known infection, and subsequent linkage to its treatment. Our overarching goal was to inform the WHO 2020 HIV Service Delivery Guidelines update.”

3. Do you used theoretical frameworks in this study? if so, please state it.

We did not utilize a specific theoretical framework. We have clarified our use of narrative synthesis methods as follows: “an approach to the systematic review and synthesis of findings from multiple studies that relies primarily on the use of words and text to summarise and explain the findings of the synthesis.”

Methods

1. Line 97: PRISMA and SWiM are stands from? Why do you use both PRIMS and SWiM? Please provide explanation. You may also state the study design properly. 

We used PRISMA guidelines for the overarching review methodology and adhered to SWiM guidelines for synthesis without meta-analysis. We have defined these terms and clarified the study design: “This multi-method systematic review and meta-analysis adhered to PRISMA (Preferred Reporting Items for Systematic Reviews and Meta-Analyses) and SWiM (Synthesis Without Meta-Analysis) reporting guidelines (S1 File: PRISMA Checklist).”

2. Line 114: Please add the information about Boolean operator used in the study. Besides, I have seen the supp file 2. Please make a separate information about searching strategy each database. Line 115: what do you mean “/” is it stands for “OR”? please clarify. 

Thank you, the forward slash meant “or” and has now been clarified : “The search strategy used the following key terms and their appropriate synonyms: 1) tuberculosis, AND 2) human immunodeficiency virus (HIV), AND diagnosis, or detection, or screening, or testing, or referral, or linkage, or coordination, or integration, or treatment initiation. The full search strategy can be found in S2 File: Search Strategy.”

3. On the PRISMA figure, please refer to PRISMA 2020 templates.

We have updated Figure 1 to adhere to the PRISMA 2020 template. 

4. For the quality assessment, how many authors contributed for it? 

We have clarified: “Two reviewers (AS, SL) assessed quality of all included studies…”.

Reviewer 2 

1. Do the authors have any particular reasons to use only Medline (OVID), Embase, Embase Classic? It is well known that PubMed is also a resourceful database including all the citations from MEDLINE plus NIH projects and PMC which also could be a good choice if one intends to search for literature related to TB and HIV. 

Our library search was developed by an expert librarian at McGill University, who advised that articles of interest would safely be covered by searching Medline and Embase databases. We have specified their assistance under methods: "With the assistance of medical librarians, we searched three electronic databases…”. We have also added the following limitation under discussion: “Finally, we included studies indexed in only two databases (Medline and EMBASE), and only in English, therefore potentially limiting the thoroughness of our results.”

2. An important keyword could be AIDS, which is missing. However, I believe the authors have covered that in their search strategy. 

The terms AIDS, HIV, and TB are added as keywords, and were covered in the search strategy (Supplementary file 2).

3. The objective is the linkage between TB and HIV care in LMICs countries; however, it seems that it has not been considered in search strategies and the authors search for all the available records. Moreover, how did the authors define the LMICs? Based which definition? 

Thank you; while we had included search terms to specify low- and middle-income countries in our search strategy (Supplemental File 2), we had not specified inclusion of these terms in the manuscript. We have now added under methods: “The search strategy used the following key terms and their appropriate synonyms 1) tuberculosis, AND 2) human immunodeficiency virus (HIV), AND 3) diagnosis, or detection, or screening, or testing, or referral, or linkage, or coordination, or integration, or treatment initiation, AND 4) low and middle-income countries. The full search strategy can be found in S2 File: Search Strategy.” We also define low- and middle income countries in the methods more clearly: “(GNI per capita <12,695 USD per year, as defined by the World Bank).” 

4. Line 134: The authors mentioned that the “discrepancies were resolved with a third reviewer (AD, VS)”. However, there are two names in the parenthesis. Please correct. 

We resolved discrepancies with either AD or VS, both senior authors on the review. We have added “or” between the two initials. 

5. Why did the authors choose to use fixed-effect meta-analysis and not a random-effect model? Thank you for raising this point. This was an oversight in editing. We did in fact use a random-effect model (DerSimonian and Laird) and have added under methods: “We performed random effects fixed effect meta-analyses (DerSimonian and Laird method) for studies implementing interventions that could be pooled (co-location interventions only).”

Reviewer 3

This systematic review is the first to assess the full spectrum of interventions to improve identification of HIV or TB disease among people with one known infection, and initiation of co-treatment in people with both. Based on the enormous individual and social burden of HIV, TB and HIV-TB co-infection, information of this nature is critically important. The problem/ research question is well motivated and the argument for the study has been developed in a clear manner. The introduction highlights why the study is important and defines the purpose of the work and its significance. The methodology that has been applied, provides an evidence base for the themes that have been identified. Themes are discussed with insight into possible reasons for findings - in this regard, the paper makes a meaningful contribution to evaluating interventions that can address identification and initiation of co-treatment for HIV, TB and HIV-TB co-infection. Based on statistical heterogeneity (I squared), the finding that joint services at the same facility are likely to result in the best HIV and TB outcomes provide direction for future interventions. The narrative synthesis of facilitators and barriers adds additional value.

1. The paper is well-written and comprehensive, but the abstract does not do justice to the full paper (e.g. indicate how homogeneity was defined; and refer to findings of meta-analysis). The abstract could benefit from language editing.

Thank you. The abstract has been re-written: Introduction: In support of global targets to end HIV/AIDS and tuberculosis (TB) by 2030, we reviewed interventions aiming to improve TB case-detection and anti-TB treatment among people living with HIV (PLHIV) and HIV testing and antiretroviral treatment initiation among people with TB disease in low- and middle-income countries (LMICs). 

Methods: We conducted a systematic review of comparative (quasi-)experimental interventional studies published in Medline or EMBASE between January 2003-July 2021. We performed random-effects effect meta-analyses (DerSimonian and Laird method) for interventions that were homogenous (based on intervention descriptions); for others we narratively synthesized the intervention effect. Studies were assessed using ROBINS-I, Cochrane Risk-of-Bias, and GRADE. (PROSPERO #CRD42018109629) 

Results: Of 21,516 retrieved studies, 23 were included, contributing 53 arms and 84,884 participants from 4 continents. Five interventions were analyzed: co-location of test and/or treatment services; patient education and counselling; dedicated personnel; peer support; and financial support. A majority were implemented in primary health facilities (n=22) and reported on HIV outcomes in people with TB (n=18). Service co-location had the most consistent positive effect on HIV testing and treatment initiation among people with TB, and TB case-detection among PLHIV. Other interventions were heterogenous, implemented concurrent with standard-of-care strategies and/or diverse facility-level improvements, and produced mixed effects. Operational system, human resource, and/or laboratory strengthening were noted within successful intervention. Most studies had a moderate to serious risk of bias. 

Conclusions: This review provides operational clarity on intervention models that can support early linkages between the TB and HIV care cascades. The findings have supported the World Health Organization 2020 HIV Service Delivery Guidelines update. Further research is needed to evaluate the distinct effect of education and counselling, financial support, and dedicated personnel interventions, and to explore the role of community-based, virtual, and differentiated service delivery models in addressing TB-HIV co-morbidity. 

Reviewer 4 

This is a well written manuscript. There are a few issues to revise if possible. 

1. Please justify why LMICs were reviewed and whether there were gold standards to compare with for each grouping (PICO1 and PICO2) eg from higher income countries etc.

Thank you. We limited the review to low- and middle-income countries (LMICs) where the highest burdens of TB and TB-HIV co-morbidity has been observed. We did not use higher income countries as a comparator or gold standard, as the epidemiology, healthcare infrastructure, and resource allocations are often not comparable. In the introduction we have now pointed to our reference point: “The World Health Organization (WHO) recommends offering routine HIV testing to all patients with presumptive and diagnosed TB, routine TB screening for TB symptoms of all PLHIV, and starting all patients with TB and HIV on both anti-retroviral therapy and anti-TB treatment (ATT).” 

2. For Table 1, can the authors be described fully or indicate in the key below that you are only listing the first author. 

We have added a footnote “a” to Table 1: “Only first authors are listed.”

3. Additionally, please link the papers in Table 1 to the reference list to make it easier to read.

We have updated the reference numbers in column 1 of Table 1 (Ref#) to mimic the reference list. 

Reviewer 5 

The authors have aggregated studies looking at HIV and TB co-morbidities and interventions to treat both diseases, and the nuances that come along with it, with the ultimate goal of improving patient care and treatment. They approach the literature analysis very thoroughly, and clearly held their analyses of articles to a high standard for inclusion. Their workflow was very clear, and the supplemental data was very helpful for anyone wishing to replicate their workflow. I detected essentially no grammatical errors, and the text was well organized and easy to read. Honestly, a great paper, and a breath of fresh air after slogging through a bunch of terrible article reviews.

1. My only issue is that your figure legends are missing. They just need a little extra info/description, especially Figure 2, which will set the precedence for each other figure's stats.

Thank you for this astute observation! We have added/elaborated the figure legends:

- Figure 1: PRISMA Study Selection Flow Chart. PRISMA = Preferred Reporting Items for Systematic Reviews and Meta-Analyses

- Fig 2. Meta-analysis and forest plot of the effect of co-location interventions (at the facility vs. within the same provider; for just testing, treatment initiation, or both) on outcomes of HIV testing and ART initiation for people with TB (PICO 1).

- Figure 3: Forest plot (not pooled) depicting the effect of interventions implementing co-location (only observed at the level of the facility) on outcomes of TB diagnosis and treatment initiation for people living with HIV. ATT = Anti-tuberculosis Treatment

- Figure 4: Forest plot (not pooled) depicting the effect of patient education and counselling interventions on all outcomes. *All studies implemented concurrent interventions and/or SOC strategies. Only the first author of each study is listed. ATT = Anti-tuberculosis treatment.

- Fig 5. Forest plot (not pooled) depicting the effect of dedicated personnel interventions on all outcomes. *All studies implemented concurrent interventions and/or SOC strategies. Only the first author of each study is listed. 

- Fig 6. Forest plot (not pooled) depicting the effect of patient peer support interventions on all outcomes. *All studies implemented concurrent interventions and/or SOC strategies. Only the first author of each study is listed. ATT = Anti-tuberculosis treatment.

- Fig 7. Forest plot (not pooled) depicting the effect of patient financial support interventions on all outcomes. *All studies implemented concurrent interventions and/or SOC strategies. Only the first author of each study is listed. ATT = Anti-tuberculosis treatment.

---

## [Decision Letter · Decision Letter 1]

11 Apr 2022

Interventions to improve linkage along the HIV-tuberculosis care cascades in low- and middle-income countries: a systematic review and meta-analysis

PONE-D-21-39116R1

Dear Dr. Daftary,

We’re pleased to inform you that your manuscript has been judged scientifically suitable for publication and will be formally accepted for publication once it meets all outstanding technical requirements.

Kind regards,

Gabriel O Dida, PhD

Academic Editor

PLOS ONE

Additional Editor Comments (optional):

Reviewers' comments:

Reviewer's Responses to Questions

**Comments to the Author**

1. If the authors have adequately addressed your comments raised in a previous round of review and you feel that this manuscript is now acceptable for publication, you may indicate that here to bypass the “Comments to the Author” section, enter your conflict of interest statement in the “Confidential to Editor” section, and submit your "Accept" recommendation.

Reviewer #1: All comments have been addressed

Reviewer #2: All comments have been addressed

Reviewer #5: All comments have been addressed

2. Is the manuscript technically sound, and do the data support the conclusions?

Reviewer #1: Yes

Reviewer #2: Yes

Reviewer #5: (No Response)

3. Has the statistical analysis been performed appropriately and rigorously? 

Reviewer #1: Yes

Reviewer #2: Yes

Reviewer #5: (No Response)

4. Have the authors made all data underlying the findings in their manuscript fully available?

Reviewer #1: Yes

Reviewer #2: Yes

Reviewer #5: (No Response)

5. Is the manuscript presented in an intelligible fashion and written in standard English?

Reviewer #1: Yes

Reviewer #2: Yes

Reviewer #5: (No Response)

6. Review Comments to the Author

Reviewer #1: Dear Author,

Thank you for addressing my suggestion very well. However, I have very minor revision about "PRISMA (Preferred Reporting Items for Systematic Reviews and Meta-Analyses) and SWiM (SynthesisWithout Meta-Analysis)"  Should be Preferred Reporting Items for Systematic Reviews and Meta-Analyses (PRISMA) and SynthesisWithout Meta-Analysis (SWiM). Please make sure all abbreviation is well spelled on galley proof.

Thank you

Reviewer #2: The authors fully addressed all the comments and I'd glad to recommend publication of this manuscript.

Reviewer #5: (No Response)

7. PLOS authors have the option to publish the peer review history of their article (what does this mean?). If published, this will include your full peer review and any attached files.

Reviewer #1: No

Reviewer #2: **Yes: **Omid Dadras

Reviewer #5: **Yes: **Jonathan LeCureux

---

## [Editor Report · Acceptance letter]

18 Apr 2022

PONE-D-21-39116R1 

Interventions to improve linkage along the HIV-tuberculosis care cascades in low- and middle-income countries: a systematic review and meta-analysis 

Dear Dr. Daftary:

I'm pleased to inform you that your manuscript has been deemed suitable for publication in PLOS ONE. Congratulations! Your manuscript is now with our production department. 

Kind regards, 

on behalf of

Dr. Gabriel O Dida 

Academic Editor

PLOS ONE